# Transcriptome Analysis Reveals Drought-Responsive Pathways and Key Genes of Two Oat (*Avena sativa*) Varieties

**DOI:** 10.3390/plants13020177

**Published:** 2024-01-09

**Authors:** Weiwei Xu, Laichun Guo, Chunlong Wang, Liming Wei, Qiang Wang, Qinyong Ren, Xiwu Yang, Chao Zhan, Xiaotian Liang, Junying Wang, Changzhong Ren

**Affiliations:** 1Agronomy College, Jilin Agricultural University, Changchun 130118, China; weiwei_xu126@126.com (W.X.); renqinyong1@163.com (Q.R.); 18252720510@163.com (X.Y.); liangxiaotian@mails.jlau.edu.cn (X.L.); 2National Oat Improvement Center, Baicheng Academy of Agricultural Sciences, Baicheng 137000, China; guolaichun@126.com (L.G.); 13500830529@139.com (C.W.); vv2315@126.com (L.W.); wangqiangeternity@icloud.com (Q.W.); 18204366627@163.com (C.Z.); 3College of Food and Biological Engineering, Chengdu University, Chengdu 610106, China; 4Biotechnology Research Institute, Chinese Academy of Agricultural Sciences, Beijing 100081, China

**Keywords:** oat, soil drought stress, leaf ultrastructure, cell wall

## Abstract

To cope with the yield loss caused by drought stress, new oat varieties with greater drought tolerance need to be selected. In this study, two oat varieties with different drought tolerances were selected for analysis of their phenotypes and physiological indices under moderate and severe soil drought stress. The results revealed significant differences in the degree of wilting, leaf relative water content (RWC), and SOD and CAT activity between the two oat genotypes under severe soil drought stress; moreover, the drought-tolerant variety exhibited a significant increase in the number of stomata and wax crystals on the surface of both the leaf and guard cells; additionally, the morphology of the guard cells was normal, and there was no significant disruption of the grana lamella membrane or the nuclear envelope. Furthermore, transcriptome analysis revealed that the expression of genes related to the biosynthesis of waxes and cell-wall components, as well as those of the WRKY family, significantly increased in the drought-tolerant variety. These findings suggest that several genes involved in the antioxidant pathway could improve drought tolerance in plants by regulating the increase/decrease in wax and cell-wall constituents and maintaining normal cellular water potential, as well as improving the ability of the antioxidant system to scavenge peroxides in oats.

## 1. Introduction

Drought stress is one of the most adverse abiotic stresses; it limits plant growth and productivity [1]. Drought stress causes reactive oxygen species (ROS) injury, plant stomatal closure, leaf growth inhibition, and an increase in photorespiration, which decreases the photosynthetic rate and disrupts plant growth and development. The antioxidant enzyme system, osmotic regulatory substances and secondary metabolites help minimize the adverse effects of drought [2]. Epidermal waxes and/or thick leaf mulch and cell-wall elasticity play important roles in maintaining cell membrane stability, reducing oxidative damage and improving cell antioxidant capacity [3].

The cell wall is the first anatomical barrier of defense against different adverse biotic and abiotic stresses; it is composed of many complex compounds, including cellulose, hemicellulose, pectin, lignin and cell-wall proteins [4]. Pectin is a cell-wall polysaccharide that acts as a water-absorbing structure. The increase in cell-wall pectin chains in drought-resistant wheat varieties under drought stress leads to the accumulation of water for cellular wall needs [4,5]. An increased hemicellulose content enhances cell-wall stiffness and prevents cell collapse due to dehydration [6]. Lignified cell walls function as waterproof lignin-rich barriers that confer drought tolerance by maintaining cell osmotic stability and protecting membrane integrity [7]. Cinnamoyl coenzyme A reductase (CCR) catalyzes the initial step of lignin monomer biosynthesis, and transcription factors such as OsNAC5 mediate lignin biosynthesis by activating OsCCR10, thereby contributing to drought tolerance [8]. Epidermal waxes can limit nonstomatal water loss [9]. Increased epidermal wax biosynthesis improves plant resistance to drought stress. Among the enzymes involved in the synthesis of waxes, β_ketoacyl CoA synthase (KCS) is the key enzyme for the elongation of very long-chain fatty acids (VLCFAs), which are responsible for the synthesis of wax component precursors [9,10]. Twenty-one KCS family genes have been functionally annotated in the Arabidopsis genome, most of which are associated with the biosynthesis of waxy components of the plant epidermis [10]. Studies have shown that the heterologous overexpression of MdKCS2, VvKCS12 and VvKCS14 promotes the biosynthesis and deposition of epidermal waxes in Arabidopsis leaves and enhances plant drought resistance [11,12].

The excessive damage to proteins, lipids and nucleic acids caused by drought stress leads to weak ROS scavenging, which disrupts the function of key organelles such as mitochondria and chloroplasts [4]. Several functional genes related to scavenging ROS in plants, including peroxidase synthesis products, ion transporters, phosphatases, calcium-dependent and transcription-regulated protein kinases, as well as the drought-responsive transcription factors (TFs) AREB, NAC, WRKY, AP2/ERF, MYB, bZIP and MYC, have been identified and characterized using modern genetic and functional genomics approaches. The WRKY gene family plays important roles in plant development and stress responses [13]. A number of family genes have been shown to participate in the regulation of plant drought resistance through the ROS pathway. For example, the mint McWRKY57 enhances the activities of the antioxidant enzymes catalase, superoxide dismutase and peroxidase in transgenic plants and improves drought tolerance [14]. Overexpression of TaWRKY1-2D enhanced drought tolerance in transgenic Arabidopsis and TaWRKY1-2 silencing in wheat increases the MDA content and reduces the proline and chlorophyll contents as well as antioxidant enzyme activity [15]. Both AtWRKY75 and AtWRKY44 are involved in the development of root hair. MaWRKY80 mediates stomatal movement and leaf water retention capacity by regulating the transcription of 9-cis-epoxysteroid dioxygenases (NCEDs) and the biosynthesis of ABA in Arabidopsis [16]. ZmWRKY40 enhances drought tolerance in transgenic Arabidopsis plants by modulating stress-related gene expression under drought stress, where ROS levels are reduced by enhancing the activities of peroxidase (POD) and catalase (CAT) [17].

Oat is an important grain feed crop in the arid and semi-arid regions of northern China [18,19]. Because of the growing urgency of water scarcity, the selection and breeding of more drought-resistant oat varieties is an important step for coping with the severe yield losses caused by extreme weather. In this study, based on the evaluation of 300 oat germplasm resources for drought resistance, two oat varieties with different drought tolerances are selected to reveal the molecular mechanisms of drought tolerance according to phenotype, morphology, physiology and gene expression level; these results provide a feasible way to select and breed strongly drought-resistant oat varieties.

## 2. Results

### 2.1. Avena Sativa Waoat Is More Drought Tolerant than Kwant

To investigate the drought tolerance phenotypes of Kwant and Waoat oats, three different moisture-level treatments were selected (Appendix A). Under moderate soil drought stress (T1), the phenotypes of Kwant and Waoat plants were not significantly different from those of the normal well-watered control (CK) plants, nor were they significantly different between the Kwant and Waoat plants. Under severe soil drought stress (T2), the leaves of Kwant plants became bent, drooped and wilted, while the leaves of Waoat plants remained in good condition. The leaf relative water content (RWC) did not significantly differ between the two varieties after moderate stress, and it was significantly greater than that in the CK treatment under severe stress; the RWC in Kwant was significantly lower (by 9.01%) than that in Waoat (Figure 1a). These findings indicate that, compared with Kwant, Waoat has better leaf water retention, less wilting and greater drought tolerance under severe soil drought stress.

To analyze the physiological mechanisms underlying drought resistance in the two oat varieties, several physiological indicators were tested. The total chlorophyll content of the drought-sensitive variety Kwant significantly decreased under moderate and severe drought conditions; however, the total chlorophyll content of the drought-tolerant variety Waoat was not significantly different from that of CK under soil drought stress (Figure 1b). The MDA content and SOD, POD and CAT activities were greater in the drought-treated groups than in the CK groups (Figure 1c–f). The MDA content of the two varieties gradually increased under soil drought stress compared with that in the CK treatment. Similarly, the MDA content increased by 15.77% in Kwant and by 10.81% in Waoat, and the increase in the MDA content was greater in Kwant leaves than in Waoat leaves (Figure 1c). Compared to those in the CK treatment, the activities of SOD, POD and CAT in Kwant plants were significantly greater after moderate soil drought stress, but there were no significant differences under severe soil drought stress (Figure 1d–f). Waoat had significantly higher levels of SOD and CAT activities under moderate and severe soil drought stress. The levels of SOD, POD and CAT activities in Waoat leaves were 23.98%, 4.15% and 12.70% higher than those in Kwant leaves, respectively, indicating that Kwant could tolerate moderate drought (60% field-water holding capacity [FWC]), while Waoat could tolerate severe drought (45% FWC). These physiological indices were consistent with the phenotypes of the two plant varieties under soil drought stress, indicating that Waoat has a stronger drought tolerance than Kwant.

### 2.2. Leaf Ultrastructure Analysis under Soil Drought Stress

To further study the structural changes in the oat leaf under soil drought stress, the anatomical structures of the leaf were observed by scanning electron microscopy (SEM) (Figure 2). The results showed that, compared with those under the CK treatment, the number of stomata increased and the stomatal area decreased in the Kwant and Waoat plants under moderate soil drought stress (Appendix A), and the morphology of the stomatal guard cells was not affected. After severe soil drought stress, the leaf surfaces in Kwant were partially damaged, the wax crystals were densely accumulated, the stomatal area decreased significantly, the stomatal number increased insignificantly and the stomata guard cells were significantly wrinkled. In Waoat plants, the leaf surfaces were undamaged, the stomatal number increased significantly (Appendix A), stomata guard cells were full and showed normal morphologies and the stomata guard cells were similarly covered with wax crystals. Further observations were performed using transmission electron microscopy (TEM) on mesophyll cells. Compared to those in CK, some disruption of the cell membrane structure in oat (Kwant and Waoat) leaf occurred after moderate soil drought stress; meanwhile, both the grana lamella and the nuclear envelope were severely disrupted in the drought-sensitive Kwant variety under severe soil drought stress, but the nuclear envelope was partially damaged and the grana lamella was clear in the drought-resistant variety Waoat, indicating more serious damage in Kwant than in Waoat (Appendix A). These results indicate that changes in the morphological structure of leaf cells under soil drought stress are important morphological indicators of differences in drought tolerance in oats.

### 2.3. Transcriptome Profiles

#### 2.3.1. Transcriptome Sequencing

To explore the potential molecular mechanisms underlying the differences in drought tolerance between Kwant and Waoat, a comparative transcriptome analysis was performed on the leaf of Kwant and Waoat plants exposed to soil drought stress. After filtering out low-quality sequences and adapters, 32.44 to 44.17 million kb clean reads were obtained per sample. The amount of clean data for all the samples reached 9.56 Gb, with a percentage of Q30 bases of 94.48% or greater. The clean reads were subsequently mapped to the OT3098 genome assembly, and 83.55–87.15% of the total reads were mapped to the genome (Table 1). The transcripts were assembled, and a total of 168,400 genes were generated by RNA sequencing.

To evaluate the consistency between the four biological replicates of each treatment, a principal component analysis (PCA) was performed (Figure 3). The results showed that the different treatments of the two oats were clustered together individually, and the four biological replicates of each group were clustered together, indicating consistency of the biological replicates. Different gene expression patterns were found between the two oat genotypes under soil drought stress.

#### 2.3.2. Analysis of the Response of DEGs to Soil Drought Stress

DEGs involved in drought stress were screened by comparing KwantCK vs. KwantT1, KwantCK vs. KwantT2, WaoatCK vs. WaoatT1 and WaoatCK vs. WaoatT2 (Figure 4a). As shown in Figure 4a, more DEGs were shared by the same material under different drought treatments than by different varieties under the same stress treatment. After moderate drought stress treatment, 2915 DEGs were identified in Kwant, 1469 of which were upregulated and 1446 of which were downregulated. In Waoat, 2864 DEGs were enriched, 1449 were upregulated, and 1415 DEGs were downregulated. Under severe drought stress, 47.43% (6440) of the DEGs were upregulated and 52.57% (7139) were downregulated in Kwant, while 48.64% (5711) were upregulated and 51.36% (6030) were downregulated in Waoat (Figure 4b). A comparison of KwantCK vs. KwantT2 and WaoatCK vs. WaoatT2 revealed that 4444 genes were significantly expressed in Waoat and that 6282 genes were also significantly expressed in Kwant (Figure 4c); these DEGs may be associated with differences in drought resistance in different oat varieties.

#### 2.3.3. GO and KEGG Pathway Analysis of DEGs

In the KwantCK vs. KwantT2 and WaoatCK vs. WaoatT2 comparisons, the top enriched GO terms were associated with the chloroplast thylakoid membrane, transcription factor activity and oxidoreductase activity (Figure 5a,b). These results suggest that oat seedling leaves respond to soil drought stress mainly through the regulation of chloroplast membrane structure, oxidoreductase activity and transcription factor activity. Furthermore, we analyzed the functional annotation of DEGs in both oat varieties after drought treatment. In total, 11,978 DEGs were identified in Kwant and Waoat, and 8542 DEGs in Kwant and 7418 DEGs in Waoat were assigned to GO classifications. In the KwantCK vs. KwantT2 and WaoatCK vs. WaoatT2 comparisons, the top enriched GO terms were associated with the chloroplast thylakoid membrane, transcription factor activity and oxidoreductase activity (Figure 5a,b). Protein kinase activity, ADP binding and protein serine/threonine kinase activity were the top enriched GO terms in the KwantT2 vs. WaoatT2 comparison (Figure 5c). The functions of the Waoat-specific DEGs were mainly enriched in protein kinase activity and transcription factor activity (Figure 5d). Kwant-specific DEGs were mainly enriched in binding and catalytic activity.

KEGG analysis of the DEGs between each group showed that starch and sucrose metabolism, photosynthesis-antenna proteins and peroxisomes were enriched in both Kwant and Waoat under severe soil drought stress (Figure 5e,f); Waoat-specific DEGs were enriched mainly in pentose and glucuronate interconversions and the fatty acid elongation pathway (Figure 5h); Waoat-specific DEGs were enriched mainly in starch and sucrose metabolism, pyruvate metabolism and the peroxisome pathway.

### 2.4. Screening of Candidate Genes

After DEG analysis in combination with the Gene Ontology (GO) and Kyoto Encyclopedia of Genes and Genomes (KEGG) analyses, the DEGs were screened for transcription factor activity, pentose and glucuronide interconversion pathways, phenylpropanoid biosynthesis and the fatty acid elongation pathway as candidate genes for regulating oat drought tolerance.

#### 2.4.1. Related Transcription Factors Responding to Soil Drought Stress

Under severe soil drought stress, 453 differentially expressed TFs (235 up- and 218 downregulated) were identified in Kwant, and 472 (269 up- and 203 downregulated) differentially expressed TFs were identified in Waoat (Figure 6b). These TFs belonged mainly to the NAC (NAM/ATAF/CUC), bHLH (basic helix-loop-helix), MYB-related (v-myb avian myeloblastosis viral oncogene homolog), bZIP (basic region-leucine zipper) and WRKY (WRKY protein) families. Compared with those in KwantT2 vs. WaoatT2, the expression levels of 287 TFs were altered, and NAC, WRKY and bZIP were highly increased among the Waoat-upregulated DEGs (Figure 6c). According to the KwantCK/KwantT2 and WaoatCK/WaoatT2 comparisons, the 200 Waoat-specifically expressed TFs (Figure 6d) and the upregulated TFs were mainly enriched in the WRKY (WRKY protein), NAC and MYB-related families, and the downregulated TFs were mainly in the bHLH, MYB-related and bZIP families.

Under severe soil drought stress, there were more DEGs in Waoat than in Kwant. Twenty-seven WRKY genes were upregulated in Waoat, and two WRKY genes were upregulated in Kwant. Studies have shown that the differentially expressed WRKY TF family genes may be candidate genes causing differences in drought resistance between these two varieties.

#### 2.4.2. Regulation of Cell-Wall Constituent Synthesis/Degradation Genes under Soil Drought Stress

Polysaccharide polymers such as cellulose, hemicellulose and pectin; glycoproteins; and lignin are the main components of plant cell walls. After soil drought stress treatment, many DEGs in both oat varieties were enriched in the pentose and glucuronide interconversion pathways, including many cell-wall polysaccharide synthesis genes (UGDH) (Figure 5h and Figure 7); a total of 53 genes (36 upregulated and 17 downregulated) were expressed in Kwant, and 66 genes (46 upregulated and 20 downregulated) were expressed in Waoat. Twelve UGDHs were significantly upregulated only in Waoat, and three UGDH genes were upregulated only in Kwant. Lignin biosynthesis genes, such as 4-coumarate (4CL), cinnamoyl-CoA reductase (CCR) and cinnamyl alcohol dehydrogenase (CAD), are involved in the phenylpropanoid biosynthesis pathway. Fifteen CCR (13 upregulated and 2 downregulated) genes were found in Waoat; in Kwant, 13 CCRs (six upregulated and seven downregulated) were enriched. Moreover, six 4CL genes were upregulated in both Kwant and Waoat, with nine CAD (five upregulated and four downregulated) genes in Kwant and seven CAD (five upregulated and two downregulated) genes in Waoat. The upregulation of many CCR genes under drought stress was an important factor for drought tolerance in Waoat. The UGDH, CCR, 4CL and CAD genes were shown to be involved in the response to drought stress in both drought-sensitive and drought-tolerant varieties.

#### 2.4.3. Genes Associated with Cuticular Wax Synthesis

The fatty acid elongation pathway was significantly enriched in Waoat according to KEGG pathway analysis (Figure 5h): 30 genes were upregulated and 19 were downregulated. In Kwant, 15 genes were upregulated and 12 genes were downregulated. There were many KCS genes related to wax synthesis, including 7 KCS genes (3 upregulated) in Kwant and 15 KCS genes (7 upregulated) in Waoat. Combined with the morphological changes in the leaf ultrastructure of both oat varieties under soil drought stress, these findings suggest that KCS genes may be candidate genes involved in drought tolerance and are associated with the regulation of leaf wax. Combined with the changes in leaf ultrastructure in both Kwant and Waoat under drought stress, these KCS genes may be candidate genes that encode leaf wax enzymes associated with drought tolerance.

### 2.5. Candidate Gene Expression Pattern Analysis

We selected nine genes from the screened candidates and analyzed their gene expression under different soil drought stress levels. The results showed that the expression of two genes related to wax synthesis, Pepsico2_Contig2119 and Pepsico2_Contig9380 (3-ketoacyl-CoA synthase 11), in the drought-sensitive variety Kwant was not significantly different from that of CK under moderate and severe soil drought stress, whereas in the drought-tolerant variety Waoat, the expression of these two genes was significantly greater different from that of CK under severe stress. The cell-wall polysaccharide polymer synthesis genes UGDH (such as Pepsico1_Contig13427.path2 and Pepsico1_Contig12917.path2) were significantly (3.25-fold and 2.34-fold, respectively) upregulated in Waoat but not in Kwant under severe soil drought stress. The expressions of the Pepsico1_Contig4337 and Pepsico1_Contig6556 (Cinnamoyl-CoA reductase 1) genes in the lignin metabolic pathway were 3.36 and 2.10, 7.96 and 3.13-fold higher than that of CK under severe and moderate soil drought stress, respectively, in Waoat, while the expression of those genes in Kwant leaf was not significantly different. Avena_sativa_newGene_22462 (glutathione S-transferase) is related to antioxidants, and its expression pattern differed significantly between the two oat varieties: the expression increased 1.38-fold in Kwant, while in Waoat it was elevated 9.58-fold. There were two regulators of drought stress responses, Pepsico1_Contig4654 (WRKY30) and Pepsico1_Contig636 (calcium-binding protein CML45). Gene expression increased in Waoat and Kwant with increasing drought severity, and the expression of these two genes significantly increased in Waoat (Figure 8).

## 3. Discussion

When plants suffer from drought stress, there are many adverse effects at the morphological, physiological, biochemical and molecular levels, including a reduction in plant height, leaf wilt, osmotic imbalances, damage to the cell membrane system and a decrease in respiration and photosynthetic rate [2,20]. Phenotypic observations of drought tolerance revealed that severe soil drought stress (45% field-water holding capacity, FWC) was the moisture threshold for the presence or absence of leaf wilt in drought-resistant and drought-sensitive varieties (Appendix A). TEM revealed that the resistant variety Waoat can protect the morphological integrity of its guard cells and maintain the stability of its cell membrane system by increasing the number of stomata and the distribution of wax crystals on its leaves under severe soil drought stress, whereas the changes in leaf ultrastructure in the two oat varieties under moderate soil drought stress were small (Appendix A). These findings indicate that oats are able to withstand certain drought conditions but that there are significant differences in drought resistance among oat varieties.

Changes in the content of osmoregulators as well as antioxidant enzyme activities are important physiological indicators for assessing plant resistance to soil drought stress; meanwhile, osmoregulators play an important role in protecting cells from ROS damage and maintaining cell membrane stability [2,21]. Under drought stress, the increase in MDA content was significantly greater in the sensitive variety Kwant than in the resistant variety Waoat, whereas the activities of SOD, POD and CAT peaked under moderate stress and then declined to a level similar to that in the CK as drought intensified (Figure 1d–f). Under severe soil drought stress, the SOD, POD and CAT activities in Waoat leaves were similar to those under moderate drought conditions, which further confirmed that the cellular structure of the drought-sensitive variety Kwant was damaged and that this damage affected antioxidant enzyme activities under severe soil drought stress (Figure 1d,f and Appendix A) because there was no significant increase in the ability to scavenge peroxides, resulting in disruption of the membrane system and a significant decrease in chlorophyll content under drought stress (Figure 1b). In the resistant variety Waoat, the activities of SOD, POD and CAT increased significantly, which contributed to reducing peroxide damage to the cellular structure and membrane system and maintaining the stability of the cellular and membrane structures (Figure 1d,f and Appendix A). Similarly, the chlorophyll content decreased without significant change (Figure 1b); thus, the leaves were able to carry out normal photosynthesis, which improved the ability of the oats to resist soil drought stress. Under drought stress, the expression of the glutathione transferase gene was strongly increased in the resistant variety Waoat, but there was no significant difference in the sensitive variety Kwant (Figure 8). Previous research has shown that under severe drought treatment, drought-resistant wheat plants exhibit systematic increases in SOD, POD, CAT, APX and ascorbate-glutathione at both the whole-cell level and in mitochondria; however, drought-susceptible wheat induces excessive accumulation of H_2_O_2_, and poorer antioxidant enzyme responses result in elevated lipid peroxidation. This suggests that excess H_2_O_2_ may inhibit or downregulate antioxidant enzyme activities under severe drought treatment [22,23], which is consistent with our experimental results. Thus, the ability of enzymatic and nonenzymatic antioxidant systems to scavenge peroxide damage is one of the main traits that accounts for differences in drought resistance between different oat cultivars.

The stability of the cellular structure and membrane system is protected by not only the antioxidant system but also the cell-wall covering and cell-wall elasticity (CWI), which play a very important role in reducing water loss from the leaf and maintaining normal cellular water potential and osmotic pressure [3,24]. In the present study, there were no significant changes in leaf morphology or relative water content (RWC) between the two oat varieties under moderate soil drought stress. However, after severe soil drought stress, the RWC was significantly higher in the drought-resistant variety Waoat than in the drought-resistant variety Kwant (Figure 1 and Appendix A).

The anatomical structure of the leaves showed that moderate drought had no significant effect on stomatal number, area or guard cell fullness, nor on dense wax crystal accumulation on the leaf surface. Anatomical structure imaging revealed that there was no significant effect on stomatal number, area or guard cell fullness, nor on dense wax crystal accumulation on the leaf surface under moderate soil drought stress. Severe soil drought stress significantly affected the stomatal and guard cell morphology in the drought-sensitive variety Kwant, which resulted in leaf wilting, whereas dense wax crystal accumulation on both leaf and guard cell surfaces occurred in the drought-resistant variety Waoat, which contributed to reduced leaf wilting and guard cell crumpling due to water deficits (Figure 2). Moreover, the stomatal number significantly increased, which could partially compensate for the decrease in stomatal area caused by soil drought stress; the increase of those stomata could facilitate the uptake of water from the soil and its transport to the leaves, maintain the normal physiological function of stomata and reduce the leaf wilt coefficient (Appendix A). These results suggest that increased wax covering of the leaf under soil drought stress was an important phenotypic trait for drought stress resistance in oats [11,12].

Water stress can disrupt the integrity of the cell wall [25]. To resume growth and adapt to water stress, plant cells must regain their ability to loosen the cell wall and incorporate new polymers over an extended period of time; this helps to avoid cell-wall disruption [26,27]. For example, increased xyloglucan content in the cell walls of bundle xylem vessels is a potential feature of resistance to drought stress in some chicory varieties [28]. Expression of xyloglucan endo-transglycosylase/hydrolases (XTHs) genes AtXTH4, AtXTH9 and CiXTH29 promoted cell-wall remodeling and contributed to better plant tolerances to drought stress [26,27,28,29]. In this study, transcriptome sequencing analyses revealed that a large number of genes related to the synthesis/degradation of cell-wall constituents as well as genes regulating wax synthesis were specifically enriched in both oat varieties with different drought resistances, indicating that the up- or downregulation of those genes is closely related to an increase in cell-wall composition and wax crystals. UDP-glucose dehydrogenase (UGDH; EC 1.1.1.22) is a key enzyme in the synthesis of polysaccharides in the cell wall [30]. Ectopic overexpression of UGDH4 in Arabidopsis significantly increased the contents of hemicellulose and soluble sugar [31]. In the present study, the expression patterns of UGDH genes were significantly different between the two oat varieties under drought stress. There was a significant 3.25-fold increase in UGDH4 expression in the drought-resistant variety Waoat under severe soil drought stress compared to that in the CK, suggesting that the high expression of UGDH4 facilitates the synthesis of hemicellulose and soluble sugars and that hemicellulose could reduce water evaporation from plant cells [7]. Many studies have shown that lignin content is an effective indicator of drought tolerance in maize [32]. Among the enzymes regulating lignin biosynthesis, cinnamoyl-CoA reductase (CCR) is a key enzyme in the lignin monomer biosynthetic pathway [33]. Similarly, CCR-catalyzed lignin biosynthesis plays an important role in drought tolerance in Rhododendron [34]. In the present study, the expression of CCR genes was significantly upregulated in the resistant variety Waoat (Figure 7), which suggested that there was greater lignin synthesis in the drought-resistant variety Waoat than in the drought-sensitive variety Kwant. Leaf wilting was less severe in Waoat than in Kwant, and cell-wall lignification may reduce water infiltration and transpiration in chloroplasts [35], which confirms the drought tolerance of Waoat.

Cuticle waxes are composed of VLCFAs and their derivatives, and β-ketoacyl CoA synthase (KCS) is the key rate-limiting enzyme for the synthesis of VLCFAs in plants [36]. Previous reports have shown that cabbage leaves, stems and other parts of the body of KCS gene mutant plants have no wax covering, and the KCS gene is significantly downregulated, suggesting that the expression of the KCS gene directly affects wax synthesis [37,38]. In this study, KCS genes (Pepsico2_Contig2119 and Pepsico2_Contig9380) were significantly increased by 2.81- and 2.29-fold, respectively, in Waoat plants under severe soil drought stress compared to Kwant plants, and SEM images showed a significant increase in wax crystals on both leaf surfaces and guard cells in Waoat plants under severe soil drought stress, indicating that high expression of the KCS gene could effectively promote wax biosynthesis (Figure 2f). There are several mutation phenotypes in the T-DNA-inserted mutant in rice, such as reduced growth, leaf fusion, and thinning of wax crystals, which contribute to increased susceptibility of the mutant to drought stress [39]. Thus, the tolerant variety Waoat may improve drought tolerance by increasing cuticular wax and reducing stomatal water loss [3].

Many studies have shown that the rapid response of WRKY family genes to drought stress and the rapid upregulation of their expression are beneficial for improving drought tolerance in crops [13,14]. The WRKY family genes ZmWRKY106, ZmWRKY40, McWRKY57-like, TaWRKY1-2D, IgWRKY50 and IgWRKY32 increase the activities of superoxide dismutase (SOD), peroxide dismutase and catalase (CAT) and decrease the content of ROS under drought stress, which helps to reduce the accumulation of H_2_O_2_ and MDA and improve the drought tolerance of transgenic plants [14,15,40,41]. In addition, the AtPOD1, AtCAT1 and AtSOD (Cu/Zn) genes, which are associated with the antioxidant system, were significantly upregulated in TaWRKY1-2D transgenic Arabidopsis plants under drought stress. TaWRKY1-2 silencing in wheat increases the MDA content, reduces the proline and chlorophyll contents and antioxidant enzyme activity, and downregulates the expression of antioxidant genes (TaPOD, TaCAT and TaSOD) [15]. In this study, we found that the expression of several WRKY family genes was significantly upregulated; SOD, POD and CAT activities were significantly increased; and the expression of glutathione S-transferase genes was significantly upregulated in the drought-resistant variety Waoat but not in the drought-sensitive cultivar Kwant under severe soil drought stress. These findings suggest that the WRKY gene may reduce the damage caused by soil drought stress in oat plants, by regulating the activity of the antioxidant system and maintaining the stability of the membrane system (Appendix A). Therefore, whether WRKY expression is strongly induced affects the degree of drought tolerance in oats. We hypothesized that some WRKY family genes play a key role in regulating the strength of drought tolerance in oats. Thus, the results show that differences in drought tolerance among different oat cultivar seedlings depend mainly on the increase/decrease in epidermal waxes, cell-wall components and the capacity to scavenge ROS via the antioxidant system. We believe that the genes related to the biosynthesis of waxes, cell-wall components and the WRKY transcription factor family play important roles in regulating drought tolerance in oats.

## 4. Materials and Methods

### 4.1. Plant Materials and Growth Conditions

*Avena sativa* materials were obtained from the National Oat Improvement Center in China, at the Baicheng Academy of Agricultural Sciences, Jilin Province, China. One drought-susceptible oat variety, Kwant, and one drought-tolerant oat variety, Waoat, were selected for this study. After disinfection and accelerated germination, the oat seeds were sown in plastic pots (14.5 cm × 9 cm × 6.5 cm) containing 0.90 kg of soil, which was a light black calcareous soil collected from the 0–25 cm tillage layer of the experimental plots. The pots were well watered to reach a soil moisture content of approximately 19%. There were 60 pots of 15 seeds per pot for each variety, and 3 replicates were established. After the seedlings had grown to approximately 15 cm, 8 seedlings with uniform growth were retained. Oat seedlings were irrigated with 0.5× Hoagland nutrient solution (30 mL/week) at the 2-leaf stage. Drought treatment was applied when the seedlings of both oat materials had grown three leaves. The two oat varieties were cultured in a greenhouse at 25 °C with a photoperiod of 16 h/8 h (light/dark).

### 4.2. Drought Treatment and Sampling

Before the soil drought stress treatment, each pot was well watered to reach a soil water content of approximately 19% (equivalent to 75% of the relative soil moisture content). The drought treatments according to Zhang et al. and Hong et al. [41,42] included moderate soil drought stress, maintaining a soil moisture content of approximately 15% (equivalent to 60% of the relative soil moisture content) and severe soil drought stress, maintaining a soil moisture content of approximately 11% (equivalent to 45% of the relative soil moisture content); soil water content was measured by the weighing method at the same time each day to replenish what was depleted [43]. The drought treatment ended when phenotypic traits appeared under severe soil drought stress for a total of 3 days. For transcriptome and physiological analysis, the leaves of three plants were selected from each pot of each variety; 20 pots were mixed as a group, frozen in liquid nitrogen and then kept at −80 °C until use. Transcriptome sequencing and physiological indicator analyses were subsequently performed.

### 4.3. Physiological Measurements

The MDA content was measured using the thiobarbituric acid method [44]. Superoxide dismutase (SOD) activity and peroxidase (POD) activity were visualized via nitro blue tetrazolium (NBT) and guaiacol colorimetry, respectively [45,46]. Catalase (CAT) activity was quantitated via the ultraviolet absorption method [45]. Sample leaves were collected under normal growth conditions and drought treatment, weighed to determine the fresh weight (FW) and subsequently weighed to determine the saturated fresh weight (SFW) after the leaf had been soaked in water for 8 h. Finally, the leaf were dried in an oven at 70 °C until they reached a constant dry weight (DW). The leaf relative water content (RWC) was calculated using the following equation:RWC (%) = (FW − DW)/(SFW − DW) × 100%,(1)

The total chlorophyll (Chl) content was measured via the UV-absorption method according to the methods of Porra et al. [47]:Total chlorophyll (mg/g) = [17.32 × A646 + 7.18 × A663] × V/(1000 × W)(2)

### 4.4. Transmission Electron Microscopy (TEM) Analysis and Scanning Electron Microscopy (SEM) Analysis

At the end of the soil drought treatment, square leaf samples (0.5 cm × 0.5 cm) were cut near the central vein of the second leaf of the oat plants, and this process was repeated three times. The leaf samples were rapidly fixed in 2.5% (*v*/*v*) glutaraldehyde for 4 h, rinsed in 0.1 M phosphoric acid buffer (PBS, pH 7.4), fixed in 1% osmotic acid buffer for 2 h and then washed three times in PBS. Then, gradient dehydration was performed on the leaves with ethanol solutions at different concentrations. After 8 h of immersion in Spurr resin at 70 °C, the sections were sliced with a Leica UC7 ultrathin slicer. Then, the cells were double-stained with uranium acetate and lead citrate, and the ultrastructure was observed by a Tecnai G2 20. For SEM analysis, the samples were fixed with 3% (*w*/*v*) glutaraldehyde for 3 h and rinsed in 0.1 M phosphoric acid buffer (PBS, pH 7.4). Then, gradient dehydration was performed on the leaves with ethanol solutions at different concentrations, after which the plants were allowed to dry naturally. The dried samples were observed by a Hitachi-SU8100 instrument after being sprayed with gold–palladium [48].

### 4.5. Total RNA Extraction and Sequencing

RNA was extracted using a Plant Total RNA Kit (Sigma, Saint Louis, MO, USA), and genomic DNA was removed with a DNase I digestion kit (Sigma-Aldrich, Shanghai, China). The RNA concentration and purity were measured with a NanoDrop 2000 photometer (Thermo Fisher Scientific, Wilmington, DE, USA). RNA integrity was assessed using an Agilent Bioanalyzer 2100 system (Agilent Technologies, Santa Clara, CA, USA). mRNA was isolated using oligo(dT)-attached magnetic beads and randomly fragmented in fragmentation buffer. First-strand cDNA was synthesized using random hexamers as primers, followed by second-strand synthesis using RNase H and DNA polymerase I. cDNA was purified using AMPure XP beads. Double-stranded cDNA was subjected to end repair, followed by the addition of adenosine and ligation to adapters. To select fragments in the size range of 300–400 bp, AMPure XP beads were applied. Twenty-four cDNA libraries were obtained by PCR amplification.

The library preparations were sequenced on an Illumina platform, and paired-end reads were generated. The raw data (raw reads) obtained were processed using Trimmomatic v0.39 [49], and the reads containing adapter contamination and nucleotides with low-quality scores were removed to obtain the clean reads. These clean reads were subsequently mapped to the oat reference genome sequence with HISAT2 version 2.2.1 [50]. The mapped reads were assembled and quantified using StringTie v2.1.5 [51], and the expression level of a gene or transcript was measured using fragments per kilobase of transcript per million fragments mapped (FPKM). Genes with a fold change of ≥2 and a false discovery rate (FDR) << 0.01 according to DESeq2 were considered DEGs [52].

### 4.6. Functional Annotation and Classification of DEGs

The differentially expressed genes (DEGs) were subjected to Gene Ontology (GO) enrichment analysis via the GOseq R package based on the Wallenius noncentral hypergeometric distribution [51]. KEGG (Kyoto Encyclopedia of Genes and Genomes) functional enrichment analysis of the DEGs was performed via KOBAS v3.0 software [52]. Venn diagramming, principal component analysis (PCA) and gene annotation (such as GenBank nonredundant, Protein family (Pfam), Swiss-Prot, and Karyotic Ortholog Groups (KOG) analyses) were performed using BMKCloud: www.biocloud.net (20 December 2023).

### 4.7. Data Analysis via Quantitative Real-Time PCR (RT–qPCR)

Nine differentially expressed genes (DEGs) were selected as candidate genes for expression pattern analysis. Total RNA extraction was performed according to Section 4.5, and cDNA was synthesized using an ueIris II RT-PCR System for First-Strand cDNA Synthesis (Us Everbright, Inc., Shanghai, China) according to the manufacturer’s instructions. An RT-qPCR assay was run using 100 ng of cDNA in a 25 μL reaction mixture containing Magic SYBR Mixture (CW3008M, CWBIO, Taizhou, China). The assay was conducted with the PCRmax Eco 48 system, and the cycling conditions consisted of an initial denaturation step at 95 °C for 10 min, followed by 45 cycles at 95 °C for 10 s and 52–63 °C for 30 s. Relative expression levels were calculated using the 2^−ΔΔCT^ method for nine genes [53]. The Actin gene was used as an endogenous reference for RT-qPCR. All analyses included three technical and biological replicates. Primers were designed using Primer Premier 6. Details regarding the RT-qPCR primers used are provided in Appendix A.

### 4.8. Statistical Analysis

The physiological parameter data were collected using Microsoft Excel v2021. GraphPad Prism v7 software was used for plotting heatmaps, and SigmaPlot v14.0 was used for the other plots. The data analysis was conducted with IBM SPSS v21 Statistics (SPSS, Inc., Chicago, IL, USA) using Duncan’s test. The statistical data are presented as the means ± standard deviations of three independent biological replicates (each consisting of eight plants). A *p* value of <0.05 was considered to indicate statistical significance.

## Figures and Tables

**Figure 1 plants-13-00177-f001:**
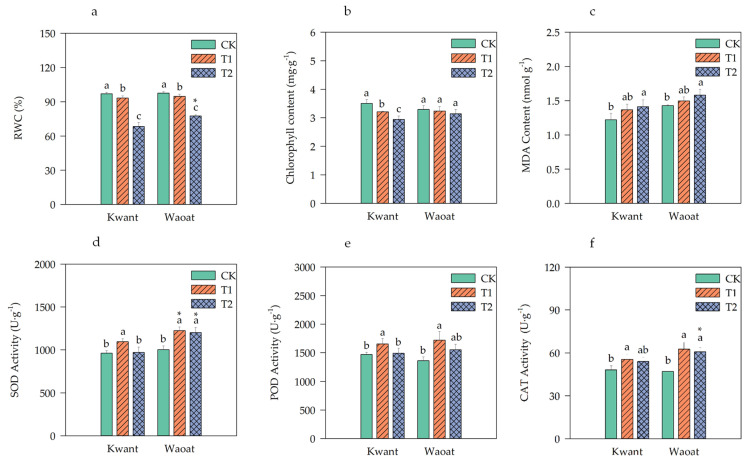
Physiological changes in Kwant and Waoat plants under different soil drought stress levels: (**a**) relative water content; (**b**) chlorophyll content; (**c**) MDA content; (**d**) SOD, superoxide dismutase; (**e**) POD, peroxidase; and (**f**) CAT, catalase. The values are presented as the means ± SDs (*n* = 3−5); three technical replicates and three biological replicates were performed. Different letters indicate significant differences within the same oat variety under different soil drought stresses (*p* < 0.05, as determined by Duncan’s test). * *p* < 0.05 indicates a significant difference within the two oat varieties under same soil drought stresses (Student’s *t* test).

**Figure 2 plants-13-00177-f002:**
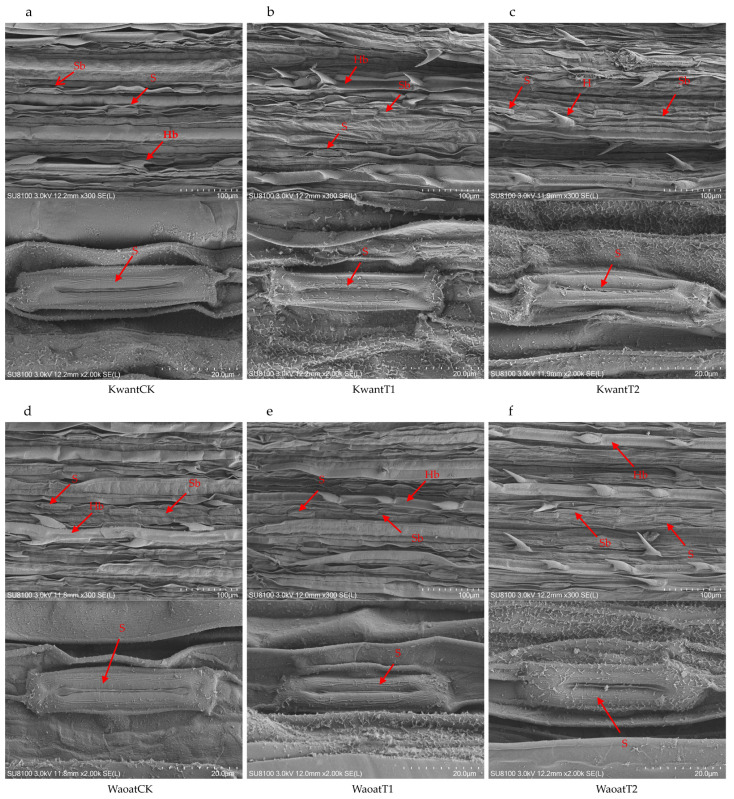
SEM analysis of the oat morphology: (**a**,**d**) leaf ultrastructure of Kwant and Waoat plants under normal moisture conditions (KwantCK, WaoatCK), (**b**) leaf ultrastructure of KwantT1 plants, (**c**) leaf ultrastructure of KwantT2 plants, (**d**) leaf ultrastructure of WaoatCK plants, (**e**) leaf ultrastructure of WaoatT1 plants, and (**f**) leaf ultrastructure of WaoatT2 plants. S—stomata, Sb—stomatal bands, H—epidermal hairs, and Hb—epidermal hair band.

**Figure 3 plants-13-00177-f003:**
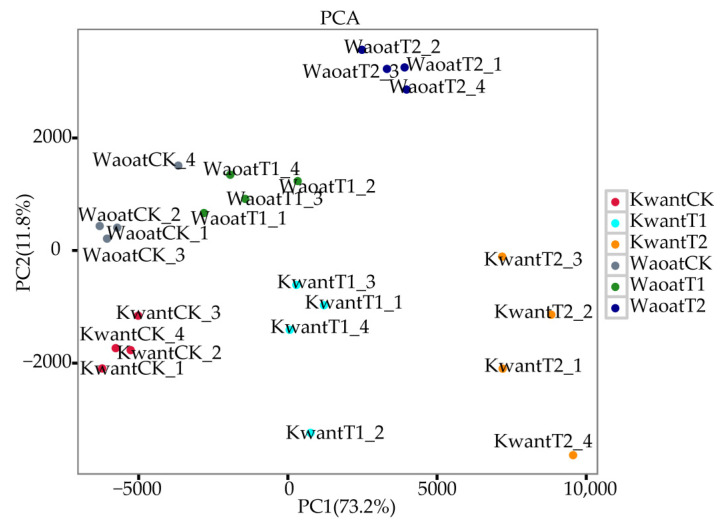
Two-dimensional PCA results of the transcriptional data from the different treatment groups.

**Figure 4 plants-13-00177-f004:**
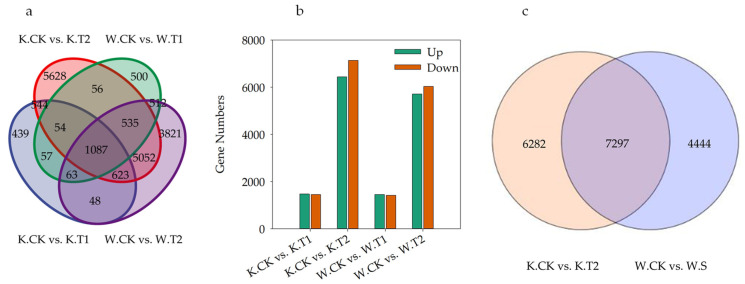
Differentially expressed genes (DEGs) between tested samples: (**a**) Venn diagram of KwantCK, KwantT1, KwantT2, WaoatCK, WaoatT1 and WaoatT2; (**b**) number of DEGs between different treatments; and (**c**) number of DEGs between KwantCK vs. KwantT2 and WaoatCK vs. WaoatT2. The upregulated DEGs in the CK vs. T2 comparison indicated that T2 exhibited more upregulated DEGs than the CK, and the downregulated DEGs showed a similar trend. K—Kwant, W—Waoat, CK—normal water control, T1—moderate drought stress, and T2—severe soil drought stress.

**Figure 5 plants-13-00177-f005:**
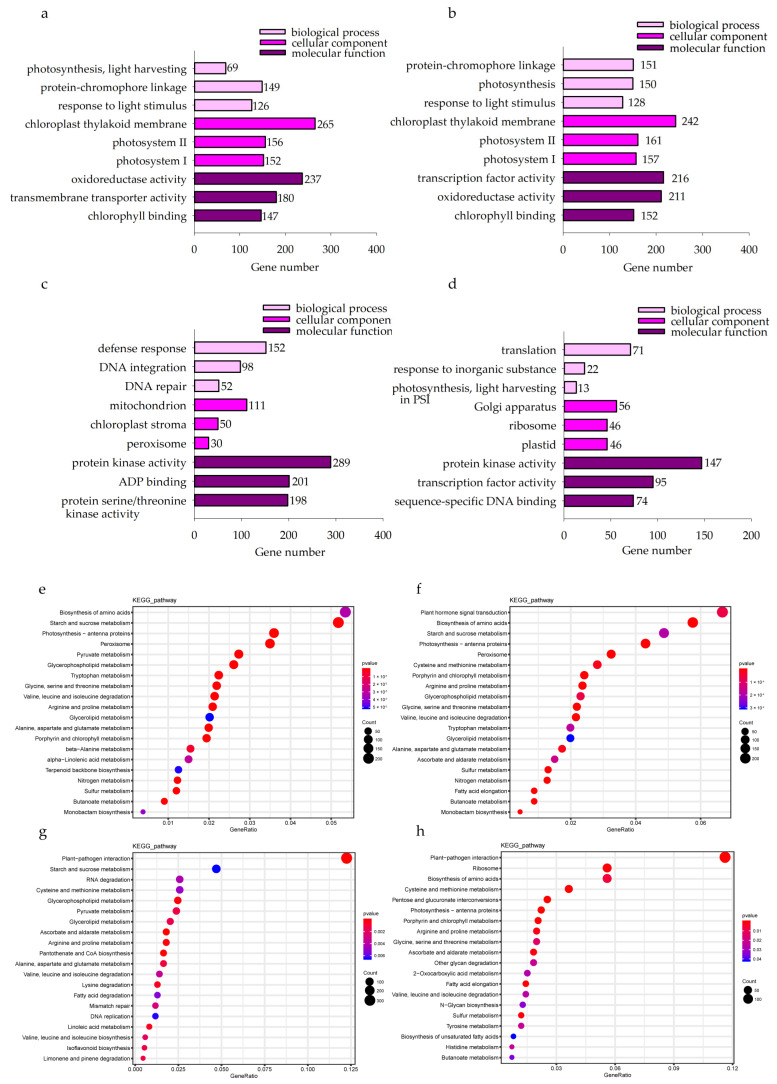
(**a**–**d**) Gene Ontology (GO) classifications of differentially expressed genes in Kwant and Waoat with different treatment (CK and T2) comparisons in the biological process, cellular components and molecular function categories. (**e**–**h**) Dot plot of the Kyoto Encyclopedia of Genes and Genomes (KEGG) pathway enrichment data for the DEGs identified in the four comparisons: (**a**) KwantCK vs. KwantT2, (**b**) WaoatCK vs. WaoatT2, (**c**) KwantT2 vs. WaoatT2, (**d**) Waoat-specifically expressed DEGs in the KwantCK vs. KwantT2 and WaoatCK vs. WaoatT2 comparisons, (**e**) KwantCK vs. KwantT2, (**f**) WaoatCK vs. WaoatT2, (**g**) KwantT2 vs. WaoatT2, and (**h**) Waoat-specifically expressed DEGs in the KwantCK vs. KwantT2 and WaoatCK vs. WaoatT2 comparisons. The size of the dot represents the number of DEGs. The different colors of the dots represent different *p*-values.

**Figure 6 plants-13-00177-f006:**
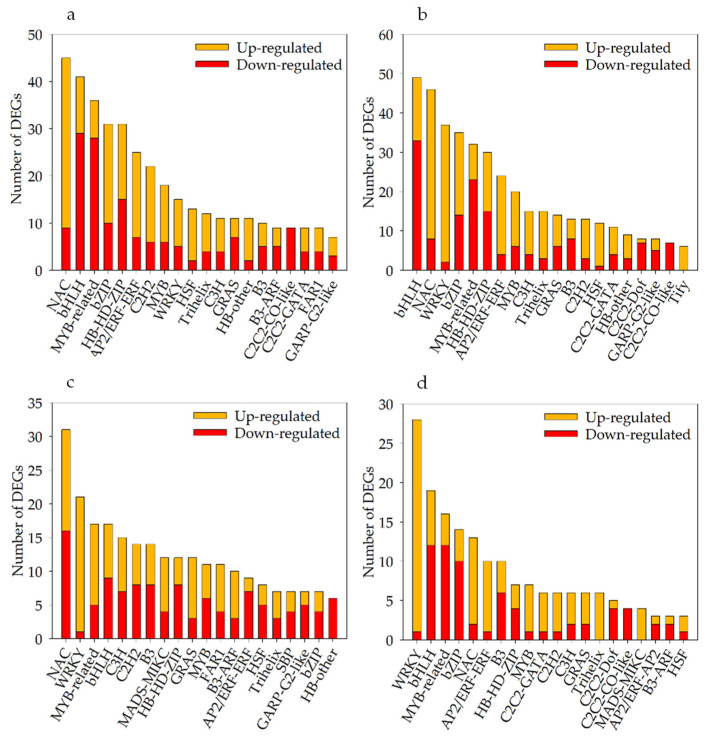
Numbers of different TF classifications among the DEGs of Kwant and Waoat: (**a**) number of up- and downregulated TFs in the two samples for KwantCK vs. KwantT2, (**b**) number of up- and downregulated TFs in the two samples for WaoatCK vs. WaoatT2, (**c**) number of up- and downregulated TFs in the two samples for KwantT2 vs. WaoatT2, and (**d**) Waoat-specifically expressed DEGs in the comparisons KwantCK vs. KwantT2 and WaoatCK vs. WaoatT2.

**Figure 7 plants-13-00177-f007:**
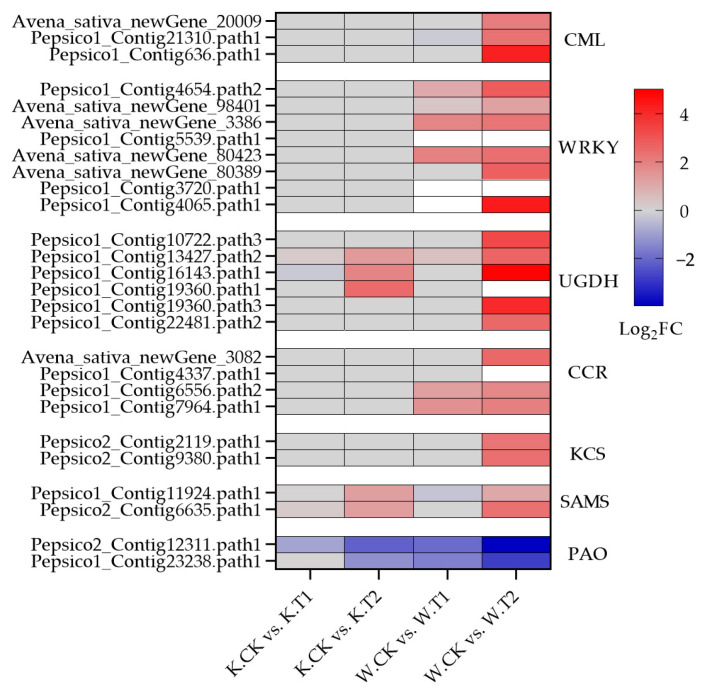
Heatmap of co-expressed soil drought stress-related DEGs in Kwant and Waoat.

**Figure 8 plants-13-00177-f008:**
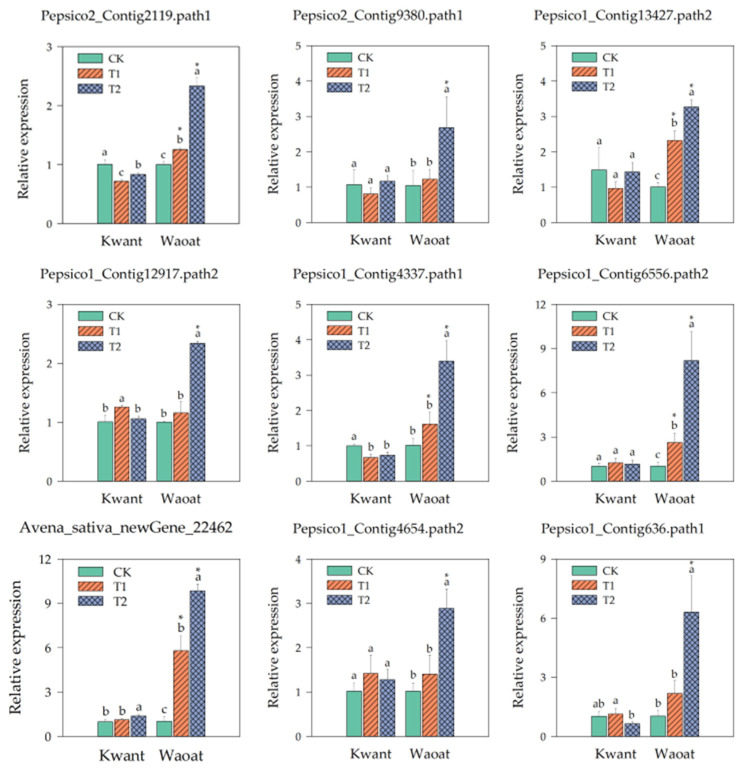
Expression analysis of selected genes after soil drought stress. RT-qPCR values were determined using the 2^−ΔΔCT^ method, with the vertical line on the column indicating the standard error of each mean. The *X*-axis indicates two oat genotypes, and the *Y*-axis indicates the relative expression of the genes. Different letters indicate significant differences within the same oat variety under different soil drought stresses (*p* < 0.05, determined by Duncan’s test). * *p* < 0.05 indicates a significant difference within the two oat varieties under same soil drought stresses (Student’s *t* test).

**Table 1 plants-13-00177-t001:** Sequencing read length data for 24 RNA sequencing libraries.

Treatment	Clean Reads	Mapped Reads (%)	GC Content (%)	% ≥ Q30
KwantCK	36,192,907 ± 992,323	85.59 ± 0.56	50.79 ± 0.46	94.78 ± 0.25
KwantT1	36,879,436 ± 574,478	87.15 ± 0.59	50.15 ± 0.64	94.88 ± 0.12
KwantT2	39,116,235 ± 3,436,238	87.01 ± 3.94	50.09 ± 0.48	94.94 ± 0.24
WaoatCK	36,983,865 ± 1,570,067	83.55 ± 4.25	50.67 ± 0.37	94.91 ± 0.16
WaoatT1	36,434,443 ± 2,818,522	86.18 ± 1.08	50.00 ± 0.19	94.93 ± 0.12
WaoatT2	37,721,158 ± 1,473,592	86.19 ± 5.26	49.47 ± 0.28	94.96 ± 0.17

## Data Availability

The genome assemblies and sequence data for *Avena sativa* Kwant and Waoat were deposited at NCBI under BioProject code PRJNA973722 (accessed on 1 December 2023). The other data generated in the study were included in this article and its Appendix A.

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
