# Peer review of "Transcriptome Analysis Reveals Drought-Responsive Pathways and Key Genes of Two Oat (Avena sativa) Varieties"

_plants, 2024, doi:10.3390/plants13020177_

Round 1

Reviewer 1 Report

Comments and Suggestions for Authors

The Manuscript ID: plants-2808760 provides a valid contribution in the search for plant species resistant to drought stress, now of wide interest in the scientific community due to ongoing global warming. The selection of two oat species differently tolerant to drought stress is well described and demonstrated through macroscopic and microscopic analyses, well supported by the transcriptomic analyzes conducted.

The manuscript is well-written, the detailed presentation of results effectively supports the findings outlined in the abstract and conclusions. However there is some inaccuracies that I have indicated in the file. In the abstract, the authors highlight an aspect at the level of the nuclear membrane and thylakoid membranes that I cannot find, as highlighted in the file.

It is also a little complicated to follow all the molecular analyzes conducted, where it is possible, I suggest specifying the acronyms of the identified genes for greater clarity.

Furthermore, the role of some genes involved in cell wall remodelling has been highlight recently in other plant species. Addressing this aspect by adding more related recent studies (https://doi.org/10.3390/biology12030444) could add valuable depth to your paper.

Author Response

Response Letter

Dear Reviewers:

Thank you for your comments and those of the reviewers, which have helped us to improve the quality of our manuscript. We have revised the manuscript and responded to each of the comments as detailed below. These changes will not the content and framework of the paper. All changes from the original manuscript are indicated in red in our revised manuscript. We have reviewed and adjusted the literature to reduce self-citation rate of our manuscript less than 15%. We hope that the changes made to the manuscript and our explanations are adequate and clear, and the manuscript is now suitable for publication in Plants. We will remain fully cooperative if you have any further comments. Thanks again.

sincerely,

JunYing Wang

Corresponding auther:

Name: Junying Wang

E-mail: wangjunying@caas.cn

And Changzhong Ren

E-mail: renchangzhong@163.com

Reviewer #1:

1) The manuscript is well-written, the detailed presentation of results effectively supports the findings outlined in the abstract and conclusions. However, there is some inaccuracies that I have indicated in the file. In the abstract, the authors highlight an aspect at the level of the nuclear membrane and thylakoid membranes that I cannot find, as highlighted in the file.

Answer

Thank you for the comments. We have revised those inaccuracies that reviewer has indicated in the file. All changes from the original manuscript are indicated in red in our revised manuscript. And then, the descriptions of changes in both the grana lamella and the nuclear envelope under severe drought stress was modified for two oat varieties. Please refer to page 5, lines 154-158.

2) It is also a little complicated to follow all the molecular analyzes conducted, where it is possible, I suggest specifying the acronyms of the identified genes for greater clarity.

Answer

Thank you for the comments. Sorry for we have made a mistake in here. Moderate drought stress had been abbreviated as M and severe drought stress as S in our first manuscript. In the submitted final manuscript, the moderate drought stress was abbreviated as T1 and severe drought stress with T2. Therefore, the M and S is not abbreviation of gene name in manuscript. We have revised the all mistakes in manuscript, and the revisions are marked in red.

3) Furthermore, the role of some genes involved in cell wall remodelling has been highlight recently in other plant species. Addressing this aspect by adding more related recent studies (https://doi.org/10.3390/biology12030444) could add valuable depth to your paper.

Answer

Thank you for the comments. We have added related recent studies in discussion. Please refer to page 13, lines 400-404.

Reviewer 2 Report

Comments and Suggestions for Authors

Dear Authirs

I read with great interest the presented manuscript, the meaning of which was to search for the morphological, physiological and molecular characteristics of the response to water deficiency in plants of an oat variety resistant to a given stressor, compared to an unstable one. The big advantage of this study is that the authors not only conducted a transcriptomic analysis, as is often the case, but also provided a good physiological basis for the stress responses of the compared oat varieties to water deficiency. From my point of view, the work is thorough, the experiments were carried out correctly, and I have practically no comments on the biological part of the work. Without going into a list of the results obtained by the authors, I would like to note only certain points that raise small questions.

1. On line 466, the authors replaced the word varieties with the word species, which is obviously a typo.

2. The methodological part does not say how the authors caused the drought. If by stopping watering, then they studied the resistance of plants to water deficiency, and not to drought, since the phenomenon of drought involves the effect of not only water deficiency, but also increased air temperature. Otherwise, the authors can talk about soil drought.

3. The resistant variety responded to water deficit by increasing stomatal density. According to the authors, an increase in the number of stomata per unit of leaf surface allows for more active absorption of carbon dioxide, since under conditions of water deficiency, the fixation of carbon dioxide by plants decreases. According to the authors, this is one of the possible reasons for increased resistance. This is a very dubious idea, since with an increase in the number of stomata, the intensity of water loss by the plant due to transpiration increases, and the key factor limiting the survival of plants in conditions of water deficiency is precisely the lack of water. Such an idea must be substantiated experimentally. This requires an objective assessment of the contribution of processes of increased water loss by a resistant plant due to an increase in stomatal density and more active fixation of carbon dioxide for the productivity and survival of plants under conditions of water deficiency.

4. Transcriptomic analysis, from my point of view, made it possible to obtain a lot of interesting data on identifying differentially expressed genes in compared varieties under water deficiency, although it made a small contribution to the understanding of the molecular mechanisms of resistance of two varieties to water deficiency.

Kind regards

Author Response

Response Letter

Dear Reviewers:

Thank you for your comments and those of the reviewers, which have helped us to improve the quality of our manuscript. We have revised the manuscript and responded to each of the comments as detailed below. These changes will not the content and framework of the paper. All changes from the original manuscript are indicated in red in our revised manuscript. We have reviewed and adjusted the literature to reduce self-citation rate of our manuscript less than 15%. We hope the changes made to the manuscript and our explanations are adequate and clear, and the manuscript is now suitable for publication in Plants. We will remain fully cooperative if you have any further comments. Thanks again.

sincerely,

JunYing Wang

Corresponding auther:

Name: Junying Wang

E-mail: wangjunying@caas.cn

And Changzhong Ren

E-mail: renchangzhong@163.com

Reviewer #2:

1) On line 466, the authors replaced the word varieties with the word species, which is obviously a typo.

Answer

Thank you for the comments. We have revised the manuscript accordingly. Please refer to Page 14, line 477.

2) The methodological part does not say how the authors caused the drought. If by stopping watering, then they studied the resistance of plants to water deficiency, and not to drought, since the phenomenon of drought involves the effect of not only water deficiency, but also increased air temperature. Otherwise, the authors can talk about soil drought.

Answer

Thank you for the comments. The drought treatments in the manuscript were carried out through controlled watering. We have revised the manuscript accordingly and replaced drought stress with soil drought stress, and in red font in the revised manuscript.

3) The resistant variety responded to water deficit by increasing stomatal density. According to the authors, an increase in the number of stomata per unit of leaf surface allows for more active absorption of carbon dioxide, since under conditions of water deficiency, the fixation of carbon dioxide by plants decreases. According to the authors, this is one of the possible reasons for increased resistance. This is a very dubious idea, since with an increase in the number of stomata, the intensity of water loss by the plant due to transpiration increases, and the key factor limiting the survival of plants in conditions of water deficiency is precisely the lack of water. Such an idea must be substantiated experimentally. This requires an objective assessment of the contribution of processes of increased water loss by a resistant plant due to an increase in stomatal density and more active fixation of carbon dioxide for the productivity and survival of plants under conditions of water deficiency.

Answer

Thank you for the comments. We agree that with an increase in the number of stomata, the intensity of water loss by the plant due to transpiration increases, however, transpiration is also the main driver for plant water uptake from the soil and transport to the ground. The reduction in stomatal area under soil drought stress reduces water loss due to transpiration on the one hand, but also affects the plant root system to take up water from the deeper soil and to transport water for the above-ground of plant, which would lead to leaves being in a more dehydrated state after the stomatal closure [1]. Whereas an increase in the number of stomata will compensate for the reduction of transpiration and photosynthesis due to the increase the degree of stomata closure [2-4]. Our results showed that both oat varieties exhibited an increase in the degree of stomata closure and stomatal number under soil drought stress, suggesting that changes in stomatal number and the degree of stomatal closure (stomatal area) was a strategy for adaptation to drought stress in oats. The higher number of stomata in drought-tolerant varieties Waoat compare to sensitive varieties Kwant, which is favorable for the absorb water from the drought soil and transported to the leaves, whereas the water-holding capacity of leaf cells is not only depend on the number of stomata and closure, the content of osmoregulatory, the cell wall covering and cell wall elasticity (CWI)played an equally important role in reducing the water loss from the leaves. We elaborated these points in the text , please refer to Page 12-13, lines 374-396, page 13, lines 421-426, page 14, lines 433-441. Therefore, we believe that the increase in stomatal number in resistant varieties under drought stress is beneficial for plant absorbed water from soil and transportation to the leaves (we have revised the manuscript accordingly, please refer to lines 29-30 in red in abstract, Page 13, line 393), while the maintenance of water potential in the leaves is achieved mainly through cell wall constituents and the increase of coverings.

References

[1]   Lawson, T.; Blatt, M.R., Stomatal size, speed, and responsiveness impact on photosynthesis and water use efficiency. Plant Physiol. 2014, 164, (4), 1556-1570. DOI: 10.1104/pp.114.237107

[2]   Caine, R. S.; Harrison, E. L.; Sloan, J.; Flis, P. M.; Fischer, S.; Khan, M. S.; Nguyen, P. T.; Nguyen, L. T.; Gray, J. E.; Croft, H., The influences of stomatal size and density on rice abiotic stress resilience. New Phytol. 2023, 237, (6), 2180-2195. DOI:10.1111/nph.18704

[3] Zhao, W.; Sun, Y.; Kjelgren, R.; Liu, X., Response of stomatal density and bound gas exchange in leaves of maize to soil water deficit. Acta Physiologiae Plantarum. 2014, 37, (1), 1704. DOI:10.1007/s11738-014-1704-8

[4] Sun, Y.; Yan, F.; Cui, X.; Liu, F., Plasticity in stomatal size and density of potato leaves under different irrigation and phosphorus regimes. Journal of Plant Physiology. 2014, 171, 1248-1255. DOI:10.1016/j.jplph.2014.06.002

4) Transcriptomic analysis, from my point of view, made it possible to obtain a lot of interesting data on identifying differentially expressed genes in compared varieties under water deficiency, although it made a small contribution to the understanding of the molecular mechanisms of resistance of two varieties to water deficiency.

Answer

Thank you for the comments. Transcription analyses do have the potential to generate many interesting data for identifying differentially expressed genes in different oat varieties under soil drought stress, however, in this paper, we focus on investigate the candidate genes related to phenotypic, morphological structures and physiological indices between two oat varieties. Other transcriptome data will be researched on next step.

Round 2

Reviewer 1 Report

Comments and Suggestions for Authors

I thank the authors that have responded point by point to my comments.